# Lassa viral dynamics in non-human primates treated with favipiravir or ribavirin

**Guillaume Lingas**[1]*, **Kyle Rosenke**[2], **David Safronetz**[3,4], **Jérémie Guedj**[1]

**1** Université de Paris, IAME, INSERM, Paris, France, **2** Laboratory of Virology, National Institute of Allergy and Infectious Diseases, National Institutes of Health, Rocky Mountain Laboratories, Hamilton, Montana, USA, **3** Department of Medical Microbiology, University of Manitoba, Winnipeg, Manitoba, Canada, **4** Zoonotic Diseases and Special Pathogens, Public Health Agency of Canada, Winnipeg, Manitoba, Canada

* guillaume.lingas@inserm.fr

**Data Availability Statement:** Data is available at FigShare, at this DOI: 10.6084/m9.figshare.13061501 (https://figshare.com/search?q=10.6084%2Fm9.figshare.13061501). The model used in Monolix 2018R2 software to fit the data using

## Abstract

Lassa fever is an haemorrhagic fever caused by Lassa virus (LASV). There is no vaccine approved against LASV and the only recommended antiviral treatment relies on ribavirin, despite limited evidence of efficacy. Recently, the nucleotide analogue favipiravir showed a high antiviral efficacy, with 100% survival obtained in an otherwise fully lethal non-human primate (NHP) model of Lassa fever. However the mechanism of action of the drug is not known and the absence of pharmacokinetic data limits the translation of these results to the human setting. Here we aimed to better understand the antiviral effect of favipiravir by developing the first mathematical model recapitulating Lassa viral dynamics and treatment. We analyzed the viral dynamics in 24 NHPs left untreated or treated with ribavirin or favipiravir, and we put the results in perspective with those obtained with the same drugs in the context of Ebola infection. Our model estimates favipiravir $EC_{50}$ *in vivo* to 2.89 $\mu$g.mL$^{-1}$, which is much lower than what was found against Ebola virus. The main mechanism of action of favipiravir was to decrease virus infectivity, with an efficacy of 91% at the highest dose. Based on our knowledge acquired on the drug pharmacokinetics in humans, our model predicts that favipiravir doses larger than 1200 mg twice a day should have the capability to strongly reduce the production infectious virus and provide a milestone towards a future use in humans.

## Author summary

Lassa virus is the etiological agent of Lassa fever, an haemorrhagic fever endemic that cause nearly 5000 deaths every year in West Africa. Here, we provide the first within-host mathematical model of the infection by Lassa virus in a macaque model and we analyze the antiviral effect of two candidate drugs, favipiravir or ribavirin. We show that both drugs act primarily by increasing mutagenesis in vivo. Both drugs had a strong antiviral efficacy in reducing the proportion of non infectious virus produced by infected cells, up to 91 and 40% for favipiravir and ribavirin, respectively. We bridge these predictions with our knowledge of the drugs pharmacokinetics to identify target concentrations having the capability to rapidly clear infectious virus from infected individuals.

non-linear mixed effect model is available at FigShare, using the following DOI: 10.6084/m9. figshare.13061588 (https://figshare.com/search? q=10.6084%2Fm9.figshare.13061588).

**Funding:** The authors received no specific funding for this work.

**Competing interests:** The authors have declared that no competing interests exist.

## Introduction

Lassa Virus (LASV; family *Arenaviridae*, genus *Mammarenavirus*) is the etiological agent of Lassa Fever (LF), a severe haemorrhagic fever. LF is endemic in West Africa, with 100,000-300,000 individuals infected with LASV every year resulting in an estimated 5,000 deaths [1] due to haemorrhage and multi-organ failure, while about 20% of survivors present long-term complications, such as hearing deficit [2]. In 2015/16, a particularly severe outbreak of LF occurred in Nigeria, associated with a mortality rate exceeding 50% and several nosocomial transmission events to health care providers, posing a serious threat to public health and resulting in a state of emergency being declared [3]. Further, the circulation of LASV is extending, with human cases detected recently in previously unaffected countries such as Mali, Burkina Faso or Ghana, leading to increasingly frequent outbreaks and leaving about 200 million people at risk for the disease [4]. Therefore treatment of LF is essentially supportive. WHO recommends high doses of ribavirin administered intravenously for severe cases, as ribavirin showed efficacy in macaques [5, 6]; however there is no clear evidence of efficacy in humans and the treatment is associated with potential side effects [7].

In the last years, several research groups were involved in the evaluation of the antiviral activity of favipiravir, an anti-influenza drug approved in Japan, against a variety of emerging RNA viruses, in particular Ebola, Lassa and Marburg viruses [8, 9]. In 2018, Guedj et al. [10] showed that early administration of favipiravir was associated with a strong antiviral activity in cynomolgus macaques infected by Ebola virus, with a 50% survival obtained at daily doses of 300 mg/kg and 360 mg/kg. The same year, Rosenke et al. [11] showed 100% survival in cynomolgus macaques infected with Lassa virus and treated up to 4 days post infection using daily doses of 300 mg/kg.

Although these results obtained in the "gold standard animal model" [12] may support the evaluation of favipiravir in clinical trials, we need to better understand how favipiravir acts *in vivo* against the virus and to better characterize its dose-concentration-antiviral activity relationship. Given the difficulty to access detailed data with BSL4 agents, it is crucial to leverage information by bridging the data collected in the different experiments. For instance the experiments performed in Ebola infected macaques provided a detailed understanding of favipiravir pharmacokinetics [13]. This information can be used to calculate the drug $EC_{50}$ *in vivo* in LASV-infected animals. Likewise the comparison of viral dynamics obtained during treatment with favipiravir in different infections can be used to better understand the drug's mechanism of action. Previous results suggested that favipiravir could increase mutagenesis but also reduce specific infectivity [14–16].

Here, our goal was to reanalyze the viral dynamics in macaques infected with LASV and treated with favipiravir or ribavirin. Using our experience acquired on drug's PK/PD gained in previous experiments of macaques infected with Ebola virus [8], we aimed to build a mathematical model that could recapitulate the viral dynamics in LASV infected animals. We used this model to provide a better understanding on favipiravir and ribavirin mode of action and antiviral activity *in vivo*. We discuss how these results could be relevant for future clinical use of favipiravir in humans.

## Materials and methods

### Study design and data provided

We reanalyzed the data provided by Rosenke et al. in two successive experiments [11]. In brief the data involve *N* = 24 female cynomolgus macaques (*Macaca fascicularis*) that received either ribavirin (RBV, *N* = 8), favipiravir (FPV, *N* = 8) or were left untreated (controls, *N* = 8). Each animal was injected intramuscularly with a lethal dose of LASV strain Josiah ($10^4$ 50% tissue

culture infective dose $TCID_{50}$). In all treated animals, treatment was initiated at D4 post-infection and was continued during 14 consecutive days. Animals treated with favipiravir received a loading dose of 300 mg/kg intravenously the first day of treatment, followed by subcutaneous injection of 50 mg/kg every 8 hours (TID, i.e., a daily dose of 150 mg/kg, $N = 4$) or 300 mg/kg once a day (QD, $N = 4$). Animals treated with ribavirin received a loading dose of 30 mg/kg the first day of treatment, followed by subcutaneous injections of 10 mg/kg TID (daily dose 30 mg/kg, $N = 4$) or 30 mg/kg QD ($N = 4$). Animals were euthanized when the clinical score was greater than 35 and surviving animals were otherwise euthanized at day 56. Animals were sampled for RNA viral load (copies/mL) and $TCID_{50}$ per mL (thereafter referred to as titers). Blood samples were collected at days 0, 3, 6, 9, 12, 15, 18, 24, 31, 42, 49 and 56, and at the time of death for euthanized animals. In 12 animals (4 controls as well as those receiving FPV TID or RBV TID), no data were available for viral titers at D3 and the qPCR limit of quantification was 2.75 $\log_{10}$ RNA copies per mL. The other animals had a qPCR limit of quantification of 3.26 $\log_{10}$ RNA copies per mL. In 4 animals (receiving RBV QD), viral titers were available only at the time of death. Limit of quantification for titers was 2 $\log_{10}$ $TCID_{50}$ per mL. For the sake of the descriptive analysis, all ribavirin treated animals were considered as a single treatment group and we analyzed the RNA and titers kinetics at the time of peak RNA viral load, comparing each treatment group to the controls.

## Viral dynamic model

Our goal was to build a within-host model of Lassa viral infection in order to unravel the host-pathogen-drug interactions and determine ribavirin and favipiravir most likely modes of action against LASV.

**Biological model.**   We used a target-cell limited model with a compartment for the innate immune response, $F$, that renders target cells, $T$, definitely refractory to infection [8]. The model includes four types of cell populations: target cells (T), refractory cells (R), infected cells in an eclipse phase ($I_1$) and productively infected cells ($I_2$). The model assumes that target cells are infected at a constant infection rate $\beta$ (mL.RNA copies$^{-1}$.day$^{-1}$). Once infected, cells enter an eclipse phase and become productively infected after a mean time $1/k$ (day). We assume that productively infected cells have a constant loss rate, noted $\delta$ (day$^{-1}$). Infected cells produce $p$ RNA copies per day (RNA copies.mL$^{-1}$.day$^{-1}$). Because both RNA viral load and infectious virus were measured, we further distinguished infectious virus, noted $V_i$, and non infectious virus, noted $V_{ni}$. We assumed that viral load, as measured by RNA copies, is the sum of infectious and non-infectious viruses, both cleared at the same rate, $c$. We assumed for simplification that 1 $TCID_{50}$ corresponds to 1 infectious virus. Infected cells produce proportionally as well a dimensionless immune effector, namely $F$, eliminated at a rate $d_f$. The model therefore can be written as:

$$\frac{dT}{dt} = -\beta V_i T - \phi T \frac{F}{F + \theta} \tag{1}$$

$$\frac{dI_1}{dt} = \beta V_i T - kI_1 \tag{2}$$

$$\frac{dI_2}{dt} = kI_1 - \delta I_2 \tag{3}$$

$$\frac{dV_i}{dt} = \frac{a}{vol}De^{-(a+c_t)t} + p\mu I_2 - cV_i \tag{4}$$

$$\frac{dV_{ni}}{dt} = p(1 - \mu)I_2 - cV_{ni} \tag{5}$$

$$\frac{dF}{dt} = I_2 - d_f F \tag{6}$$

where $\beta$ is the rate of infection of target cells and $c$ is the clearance rate of RNA copies in the circulation. Once productively infected, cells produce $p$ RNA copies per day, but only a fraction of them, $\mu$, is infectious. One can derive the basic reproduction number, $R_0 = \frac{\beta\mu p T_0}{c\delta}$ that corresponds to the numbers of cells infected by an infected cell in a population of fully susceptible cells ($T_0$). The effect of the immune response is therefore represented by the term $\phi T \frac{F}{F+\theta}$. In this model, the presence of viral antigen stimulates the innate immune response, represented by the compartment $F$, which activates and renders uninfected cells refractory to infection at a rate noted $\phi$, while $\theta$ represents the concentration of $F$ required to achieve 50% of the maximal effect.

As an initial infectious viral load is required to model the start of the infection, we used a model proposed by Best et al. [17] to model the progressive arrival of the viral inoculum to the infection site after intramuscular injection. In this model the initial inoculum dose, $D$ (equal to $10^4$ TCID$_{50}$), is diluted into the plasmatic volume, $vol$ (equal to 300 mL in macaques [18]) and is transported to the site of infection with rate $a$. During the transport, the virus can also be eliminated, with a clearance rate $c_t$.

Further, as some animals did not show any viral load rebound after treatment cessation, we modeled the possibility that animals could be cured from virus. Following what has been done in other curable viral infection (such as Hepatitis C virus), we assumed that the infection was cured if there was less than one infectious virus in the plasmatic volume, $vol$ [19]. This corresponds to having a concentration of $V_i < 0.003$ TCID$_{50}$.mL$^{-1}$.

**Favipiravir and ribavirin pharmacokinetic models.** We used published pharmacokinetic models to predict plasma drug concentrations of favipiravir and ribavirin, noted $C_{FPV}$ and $C_{RBV}$, respectively. Ribavirin pharmacokinetics was described by a one-compartment model [20] while complex favipiravir pharmacokinetics was described by a one-compartment model with an enzyme-dependent elimination rate [13] (S1 Text). Given the absence of pharmacokinetic data in the experiment, we fixed all individual parameters to the mean values found in the literature [13, 20] (see S1 Table and predictions in S1 Fig).

**Favipiravir and ribavirin modes of action.** We studied the possibility that FPV and RBV could either reduce the viral production, $p$, or decrease the proportion of infectious virus, $\mu$. We noted $\epsilon_{drug}$ the antiviral efficacy of the considered drug (i.e. $\epsilon_{FPV}$ or $\epsilon_{RBV}$) and we assumed an E$_{max}$ model to relate drug concentration to efficacy:

$$\epsilon_{FPV}(t) = \frac{C_{FPV}^\kappa(t)}{C_{FPV}^\kappa(t) + EC_{50_{FPV}}^\kappa} \tag{7}$$

$$\epsilon_{RBV}(t) = \frac{C_{RBV}(t)}{C_{RBV}(t) + EC_{50_{RBV}}} \tag{8}$$

where $EC_{50_{FPV}}$ and $EC_{50_{RBV}}$ are the drug concentrations of FPV and RBV, respectively, needed to achieve 50% efficacy. In the case of FPV, for which 2 different doses were used, we further considered the possibility of a sigmoidicity parameter (Hill coefficient, noted $\kappa$) in the concentration-effect relationship, ranging from 1 to 5 (close to an "on-off" effect). We did not consider a Hill coefficient for RBV different than 1 since only one dose was used. During

treatment the model equations given by Eqs (4) and (5) become:

$$\frac{dV_i}{dt} = \frac{a}{vol}De^{-(a+c_t)t} + p\mu(1 - \epsilon_{drug})I_2 - cV_i \tag{9}$$

$$\begin{cases} \dfrac{dV_{ni}}{dt} = p(1 - \mu)(1 - \epsilon_{drug})I_2 - cV_{ni} \\ \text{(if treatment blocks production)} \tag{10a} \\ \dfrac{dV_{ni}}{dt} = p(1 - (\mu(1 - \epsilon_{drug})))I_2 - cV_{ni} \\ \text{(if treatment makes virus non infectious)} \tag{10b} \end{cases}$$

## Parameter estimation & model building

**Assumptions on fixed parameter values.** To ensure parameter identifiability, a number of parameters had to be fixed. Viral clearance in plasma, noted $c$, respectively, was fixed to 20 d$^{-1}$, similar to what has been performed in Zika and Ebola viruses [8, 18]. For simplification, we fixed viral clearance in the tissues $c_t$ to the same value. Clearance rate of immune effector, $d_f$, was fixed to 0.4 d$^{-1}$ [8]. As only the product $pT_0$ is identifiable, we fixed $T_0$ to $10^7$ cells.mL$^{-1}$, as it is an approximation of the liver size, which is the main target of LASV [18].

**Algorithm for inference.** Parameter estimation was performed in a non-linear mixed effect model framework and the likelihood was maximized using the SAEM [21](Stochastic Approximation Expectation-Maximization) algorithm implemented in Monolix software (http://lixoft.com). Information brought by data under limit of quantification was taken into account in parameter estimation [22, 23]. Statistical criterion used for model discrimination was the BIC.

**Model building strategy.** Our model building strategy was the following (S2 Fig).

1) We considered 4 models assuming that favipiravir or ribavirin could either block viral production or viral infectivity (Eq 10a or Eq 10b). All 4 candidate models were fitted assuming different fixed values for $k$ and $a$ ranging from 4 to 20 d$^{-1}$ and 0.005 to 0.1 d$^{-1}$, respectively. At this stage, we assumed no sigmoidicity for FPV and $\kappa = 1$.

2) We considered other modes of action of the immune response, using the same parametrization that was used in previous works. We tested several scenarios where F either acted by reducing viral infectivity, decreasing the rate of viral production, increasing viral clearance or the loss rate of infected cells through a cytotoxic effect (S2 Text).

3) The selected model was then reduced to limit the number of random effects. Random effects with a standard deviation <0.1 or associated with a relative standard error of +100% were deleted by using a backward procedure and were kept out if the resulting BIC did not significantly increase by more than 2 points.

4) Finally, we aimed to account for uncertainty on the sigmoidicity parameter, $\kappa$. We considered values of $\kappa$ ranging from 1 to 5 and we estimated the model parameters in each scenario. We then used the model averaging approach proposed in Gonçalves et al. [24] to take into account model uncertainty. Thus, we calculated the weight associated to each value of $\kappa$, given by $\omega_\kappa = \frac{e^{-\frac{BIC_\kappa}{2}}}{\sum e^{-\frac{BIC_\kappa}{2}}}$. Then the parameter estimate and its confidence interval was obtained by sampling parameters in the mixture of the asymptotic distribution of the estimators of each model, with weights $\omega_\kappa$.

## Simulations studies and extrapolation

**Predicting the effects of drug human exposure.** Using the same approach of model averaging to account for model uncertainty on $\kappa$, we simulated viral dynamic profiles that could be obtained with clinically relevant residual concentrations. We assumed a fixed treatment duration of 14 days, with treatment initiation at D4, D6 or D8 post infection. We assumed constant favipiravir concentrations equal to 46.1 or 25.9 $\mu$g.mL$^{-1}$, which correspond to the mean residual concentrations obtained respectively 2 and 4 days after treatment initiation in Ebola-infected patients of the JIKI trial [25]. We also considered a larger value of 80 $\mu$g.mL$^{-1}$, which corresponds to the residual value showing efficacy in NHPs infected with Ebola virus [10]. For ribavirin, we assumed constant drug concentration of 15 $\mu$g.mL$^{-1}$, which corresponds to the mean value reported in individuals receiving the recommended dosing regimen in Lassa infection (1000 mg every 6 h) [26].

## Results

### Descriptive analysis of viral kinetics

Survival at D56 was achieved in 100% of animals receiving 300 mg/kg/day of favipiravir, whereas all other animals died within 23 days post-infection (Fig 1). Although all animals treated with ribavirin died, the survival was extended over control animals (P<0.001), showing the benefit of ribavirin. Finally, low dose favipiravir of 150 mg/kg/day showed no benefit on survival compared to untreated animals (P = 0.4).

Ribavirin and favipiravir 150 mg/kg/day had a limited effect on viral titers at peak RNA viral load with median values of 4.32 and 4.07 $\log_{10}$TCID$_{50}$.mL$^{-1}$, respectively, vs 5.90 $\log_{10}$TCID$_{50}$.mL$^{-1}$ for controls, (p<0.01 to controls in both cases). In contrast all animals receiving 300 mg/kg/day favipiravir has undetectable viral titers at all times (p<0.01 vs control, Table 1).

Although viral titers remained undetectable, high levels of viral load were observed in these animals during all the course of the infection, with peak viral load of 5.03 $\log_{10}$ RNA copies. mL$^{-1}$ compared to 7.96 $\log_{10}$ RNA copies.mL$^{-1}$ in untreated animals (p<0.01). The effect of

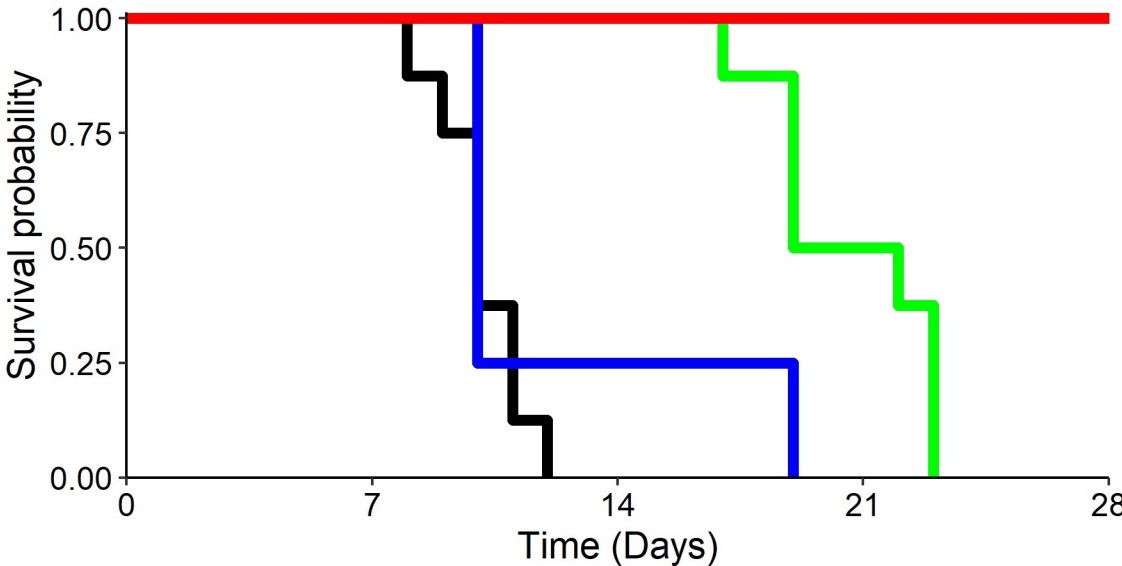

**Fig 1. Observed survival according to treatment received.** Black curve: animals receiving vehicle treatment; green: animals treated with RBV; blue: animals treated with FPV 150 mg/kg/day; red: animals treated with FPV 300 mg/kg/day.

**Table 1. Description of viral kinetics.** Median (min-max). * = p <0.05 to untreated (Log-rank test for survival, Wilcoxon test for viral load and titers).

|  | No Treatment | RBV 30 mg/kg/day | FPV 150 mg/kg/day | FPV 300 mg/kg/day |
|---|---|---|---|---|
| Time to death (d) | 10 (8-12) | 20.5* (17-23) | 10 (10-19) | NA |
| Time to peak viral load (d) | 9 (9-12) | 15* (11-15) | 10 (9-15) | 7.5 (6-15) |
| Peak viral load ($\log_{10}$ RNA copies.mL$^{-1}$) | 7.96 (7.47-8.07) | 7.68 (7.21-8.57) | 7.37* (7.20-7.94) | 5.03* (4.68-6.65) |
| Viral titers at peak viral load ($\log_{10}$TCID$_{50}$.mL$^{-1}$) | 5.90 (5.35-6.95) | 4.32* (4.20-4.44) | 4.07* (3.94-5.20) | 2* (2-2) |

favipiravir was dose dependent, with a dose of 150 mg/kg/day leading to high levels of RNA viral load at peak (7.37 $\log_{10}$ RNA copies.mL$^{-1}$, p<0.05 compared to untreated animals). In ribavirin treated animals, the peak viral load was similar to untreated animals (Table 1).

Taken together, these results suggest a greater effect of both ribavirin and favipiravir on viral titers than on viral load. To visualize that, we plotted the ratio of viral titers to viral load at the time of peak viral load, a proxy of the ratio of proportion of infectious virus (Fig 2). Interestingly there was a clear reduction of this ratio in all treated animals (p<0.05 for all groups compared to control). Of note, as viral titers were under the limit of detection in animals in animals treated with 300 mg/kg/day, the values were imputed to the limit of detection, i.e., 2 $\log_{10}$TCID$_{50}$.mL$^{-1}$. Hence this ratio represents a minimal estimate for these animals. We compared these observations with those obtained in other cynomolgus macaques infected with Ebola virus and treated with comparable doses of favipiravir [10]. Interestingly, in these animals the ratio of viral titers to viral load remained unchanged across the different doses of favipiravir and was not lower than in untreated animals (Fig 2). This suggests that the mechanism of action of favipiravir observed here against Lassa virus could be different from what was observed in Ebola infected individuals.

### Viral dynamic models

**Model building and drug's most likely mechanism of action.** The BIC obtained for all tested models are given in S3 Fig. The model assuming an effect of both favipiravir and ribavirin in reducing the proportion of infectious virus (Eq 10b for both drugs) systematically provided a better fit to the data than a model assuming that the favipiravir or ribavirin could block viral production (Eq 10a, S3A & S3D Fig, $\Delta$BIC $\approx$ 10 points in all cases considered). When considering further different levels of sigmoidicity in the concentration effect curve of favipiravir, the best model was obtained with $\kappa = 5$ (Fig 3 and S3E Fig).

We also aimed to understand in more details what were the differences in model predictions according to the putative mechanisms of action of both drugs. Assuming that both drugs block viral production not only deteriorated data fitting criterion (i.e. increased BIC by 10 points) but also provided inconsistent predictions. Indeed, the ratio of infectious virus was increased during treatment and was dose dependent, and this pattern could not be reproduced by a model assuming that treatment reduces the production of virus per infected cell (Fig 2 and S4A Fig). In contrast, assuming an effect in reducing the proportion of infectious virus could recapitulate the effect of treatment on the proportion of infectious virus (S4B Fig). Further, the analysis of the individual parameter estimates showed differences across the treatment groups in the distribution of the proportion of infectious virus, $\mu$ (S5 Fig). This is at odds with the fact that $\mu$ is not related to treatment, and therefore should be similarly distributed in all groups.

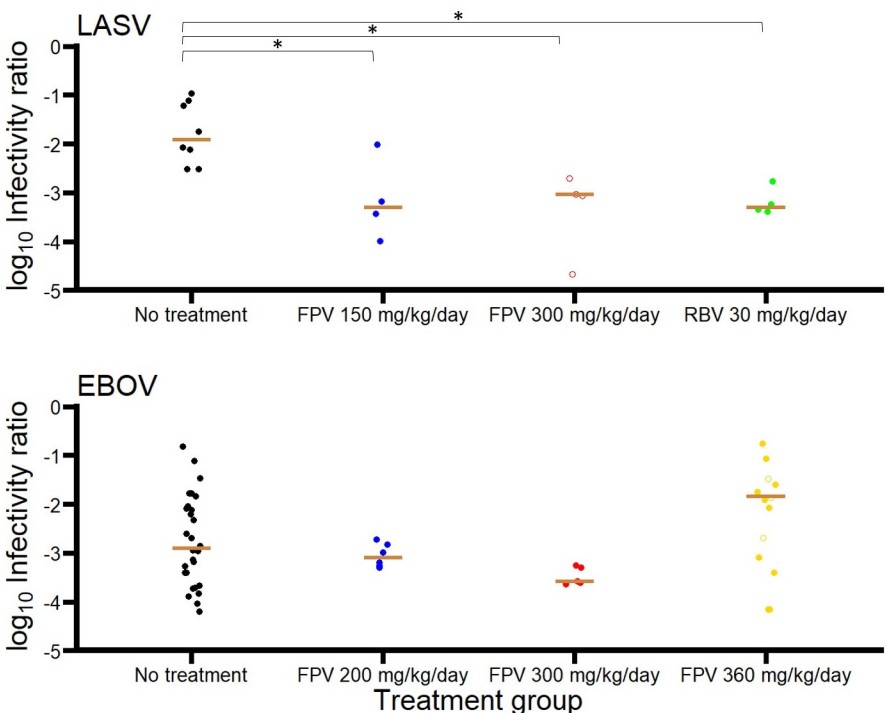

**Fig 2. Ratio of titers to viral load at the time of peak viral load in cynomolgus macaques infected with Lassa (top) or Ebola (bottom) virus.** The ratio of titers is given by $\log_{10}\left(\frac{TCID_{50}.mL^{-1}}{RNAcopies.mL^{-1}}\right)$. Animals infected with Lassa virus were either untreated (black), treated with FPV 150 mg/kg/day (blue) or 300 mg/kg/day (red), and treated with RBV (green). Similar ratio was calculated in animals infected with Ebola virus that were either untreated, treated with FPV 200 mg/kg/day (blue), FPV 300 mg/kg/day (red), or FPV 360 mg/kg/day. Empty circles correspond to undetectable viral titers imputed to the limit of detection ($2 \log_{10}TCID_{50}.mL^{-1}$). * = p<0.05 (Wilcoxon test).

**Challenging immune response's mode of action.** The refractory model outperformed all other models of immune response (S2 Text), with BIC difference of more than 9 points (S2 Table). Moreover, we verified the necessity of this immune response as well as the effector compartment. Those models showed deteriorated statistical criteria in terms of BIC, with increase of at least 6 points, supporting the use of a refractory response with an effector compartment (S3 Table).

**Parameters estimation of best model using model averaging.** Next, we took into account the uncertainty on the sigmoidicity of the concentration-effect relationship of favipiravir (see Materials and methods), using the best model determined above. Indeed, the parameter estimates obtained with varying values of $\kappa$ from 1 to 5 were largely similar, even when the concentration-effect curve was close to an on-off effect ($\kappa = 5$). Overall the model averaging results provided an estimate of the favipiravir and ribavirin $EC_{50}$ of 2.89 (95% CI 1.44-4.55) and 2.97 (95% CI 2.46-3.51) $\mu$g.mL$^{-1}$, respectively (Table 2). With the PK profile of each drug (S1 Fig), one can calculate the average antiviral efficacy during the course of treatment. At the doses of 150 and 300 mg/kg per day, favipiravir led to average efficacy of 59% and 91% with a sigmoidicity, in reducing virus infectivity. The large level of efficacy in animals treated with 300 mg/kg was sufficient to rapidly drive the infectious virus to the cure boundary after treatment initiation and reproduce extinction of infectious viruses. For ribavirin, the predicted average drug concentration was equal to 3.6 $\mu$g.mL$^{-1}$ (irrespective of the dosing regimen, see S1B Fig), corresponding to an average efficacy of 40% in reducing virus infectivity. Of note,

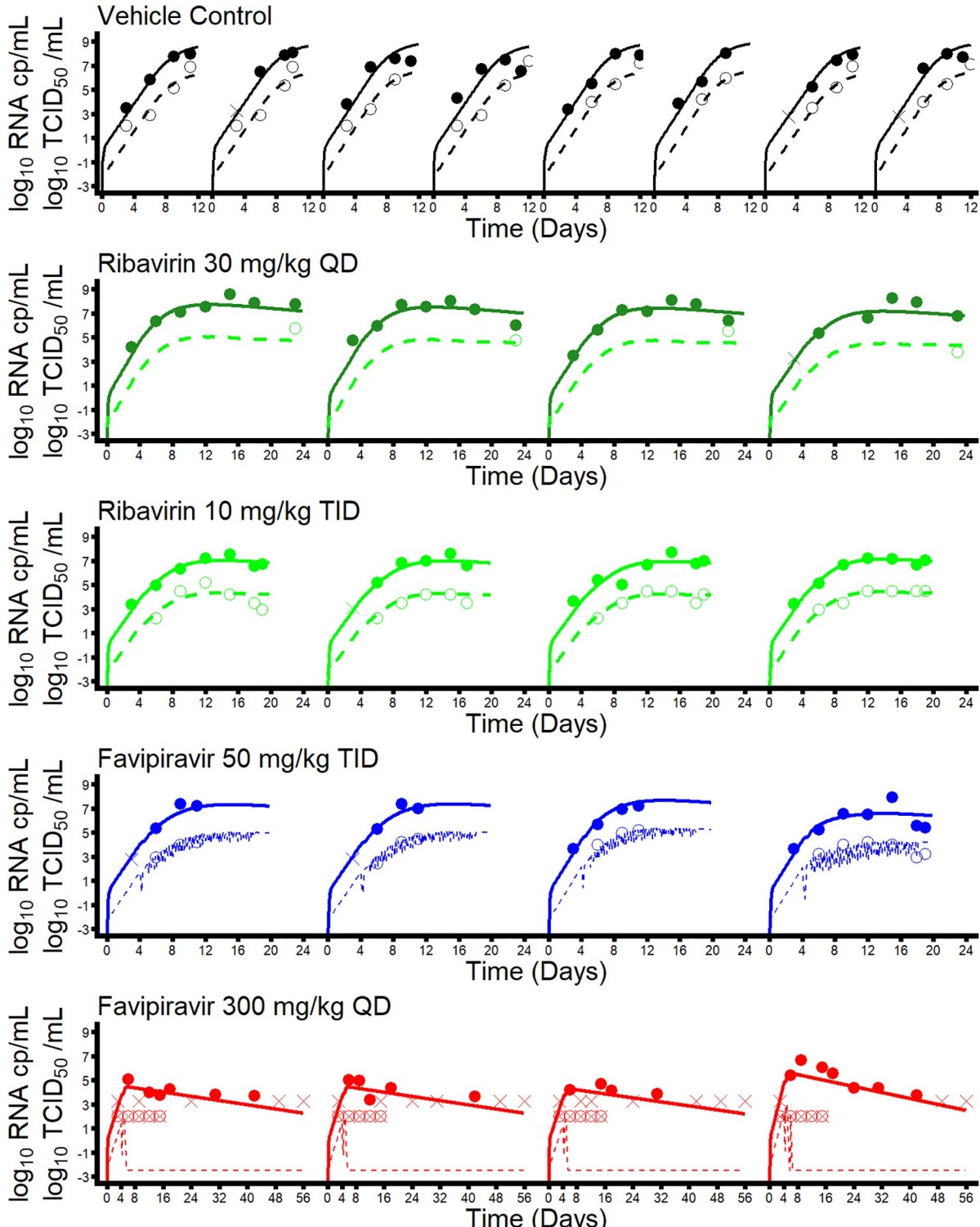

**Fig 3. Individual fits of viral load (plain lines) and viral titers (dashed line) for best model considered ($\kappa = 5$).** Plain circles represent observed viral loads and empty circles represent observed viral titers. Data below the limit of detection were represented by crosses (viral load) or crossed empty circles (viral titers).

**Table 2. Parameters distribution using model averaging (Median, 95% CI).**

| Parameter | Parameter estimates | |
| --- | --- | --- |
| | Fixed effects (Median, 95% CI) | Standard deviation of the random effect (Median, 95% CI) |
| $R_0$ | 38.6 (37.9-39.3) | 0 |
| $\phi$ (d$^{-1}$) | 0.43 (0.31-0.54) | 0 |
| $\delta$ ($10^{-1}$ d$^{-1}$) | 1.1 (1.05-1.12) | 0.13 (0.10-0.16) |
| $p$ ($10^4$ RNA copies/cell/d) | 5.86 (2.01-12.4) | 0.33 (0.01-0.65) |
| $\theta$ | 121.8 (0-393.7) | 0 |
| $EC_{50_{RBV}}$ ($\mu$g/mL) | 2.97 (2.46-3.51) | 0 |
| $EC_{50_{FPV}}$ ($\mu$g/mL) | 2.89 (1.44-4.55) | 0.95 (0.37-1.50) |
| $\mu$ | 0.005 (0.004-0.006) | 0 |
| $\sigma_{RNA}$($log_{10}$RNA copies. mL$^{-1}$) | 0.642 | - |
| $\sigma_{TCID_{50}}$ ($log_{10}$ TCID$_{50}$.mL$^{-1}$) | 0.571 | - |
| | **Fixed parameters** | |
| $a$ (d$^{-1}$) | 0.05 | - |
| $k$ (d$^{-1}$) | 4 | - |
| $c$ (d$^{-1}$) | 20 | - |
| $vol$ (mL) | 300 | - |
| $T_0$ (Cells.mL$^{-1}$) | $10^7$ | - |
| $df$ (d$^{-1}$) | 0.4 | - |
| $\kappa$ | 1-5 | - |

this estimate of EC$_{50}$ was robust to hypothesis on the mechanism of action and a model assuming an effect of both drugs in blocking viral production led to estimates of drug EC$_{50}$ of 5.82 (95% CI 2.62-9.02) and 5.1 (95% CI 3.92-6.28) $\mu$g.mL$^{-1}$ for favipiravir and ribavirin, respectively (S7 Fig).

The model included a compartment for the innate immune response, whose antigen-dependent stimulation render susceptible cells refractory to infection. We estimated that the maximal rate of conversion from susceptible to refractory, $\phi$, was equal to 0.43 d$^{-1}$ (95% CI 0.31-0.54). By depleting the compartment of susceptible cells and preventing the infection to start again by reaching the cure boundary, this mechanism explains why viral load does not increase after treatment cessation at D17 (Fig 3 and S6 Fig), even when the efficacy was modest. In all surviving animals, viral load after peak viral load declined slowly which was attributed to a loss rate of infected cells of 0.11 d$^{-1}$ (95% CI 0.105-0.112), corresponding to a half life of 6 days.

## Predictions of drug efficacy for human dosing regimens

We simulated the effect of favipiravir and ribavirin on viral dynamics for levels of drug concentrations similar to those obtained with clinical dosing regimens of favipiravir and ribavirin (Fig 4). In the JIKI trial, patients infected with Ebola virus received 1200 mg BID of favipiravir, with residual drug concentrations of 46.1 and 25.9 $\mu$g.mL$^{-1}$ respectively 2 and 4 days after treatment initiation, respectively. These dosing regimens would both lead to antiviral efficacy against Lassa infection larger than 99%. Even in the most conservative scenario where only a sigmoidicity of 1 would be considered, these levels are sufficient to achieve an efficacy greater than 90%. For ribavirin, the recommended dosing regimen against Lassa would correspond to drug concentration of 15 $\mu$g.mL$^{-1}$ (see Materials and methods), corresponding to an antiviral efficacy of 83%.

As predicted by viral kinetic models [27] the impact of these treatment is larger when it is initiated early in the infection. For instance initiating a treatment at D4 post infection (i.e., at the first detectable viral load) would lead to cure within a day with all concentrations of favipiravir considered. For treatment initiated at D6 or after, treatment may need to be administered for more than 2 weeks to drive the virus to the cure boundary. For ribavirin, the efficacy was sufficient to bend the course of virus and reduce peak viral load, but it was not sufficient to drive the virus towards extinction.

## Discussion

We developed the first mathematical model describing LASV viral dynamics during treatment with favipiravir and ribavirin in a non-human primate model of Lassa infection. The comparative approach used to discriminate between drugs modes of action showed that the best description of the data was obtained when both favipiravir and ribavirin were assumed to act reduce the proportion of infectious virus. Our model provided an estimate of the drugs $EC_{50}$, equal to 2.89 and 2.97 $\mu$g.mL$^{-1}$ for favipiravir and ribavirin, respectively. Given the viral dynamic profiles of LASV, our model predicted that favipiravir and ribavirin achieved average drug efficacy of 91 and 40% in reducing infectious virus, respectively.

One of the interest of this study was to compare with results obtained previously in the context of Ebola infection, in a similar animal model of cynomolgus macaques. The *in vivo* $EC_{50}$ identified for favipiravir against Lassa virus was much lower than against Ebola virus (2.97 $\mu$g/mL is ribavirin EC50. the correct one is Favipiravir EC50, which is 2.89 $\mu$g/mL. vs 200 $\mu$g.mL$^{-1}$, respectively [8]) and was consistent with values found in vitro of 4.6 $\mu$g.mL$^{-1}$ [28]. Another interesting observation was the difference in the drug mechanism of action in the two viral infections. Here, the data showed that favipiravir, and ribavirin to a lesser extent, had an effect in reducing the virus infectivity, as measured by the ratio of $TCID_{50}$ to RNA copies (Fig 2). In the context of Ebola and Marburg infection NHP models, favipiravir was associated with an increased virus diversity [10, 16]. A small effect on virus infectivity was reported in [16], but this was not observed in our previous study. Further studies on genomic viruses will be needed to confirm that the strong effect of favipiravir on virus infectivity is also caused by an increased in virus mutagenesis and, if so, what levels of mutagenesis might be associated with "error catastrophe". In our model, the concentration-effect relationship of favipiravir was associated with a high sigmoidicity, suggesting that error catastrophe can only be observed when the drug concentrations passes a critical threshold. Of note, mutagenesis may also increase the risk of emergence of mutations conferring resistance. Resistance to treatment were not observed in NHP models of Ebola and Marburg infections, but patterns of resistance to resistance to favipiravir were identified in in vitro models of influenza infection [29].

Another important finding of our study was the demonstration that ribavirin had an antiviral efficacy in vivo against Lassa infection. Here again the effect passed through a modulation of the infectivity rather than the viral production. In a previous mice model of Lassa infection, we identified an effect of ribavirin in reducing the loss rate of infected cells but no specific antiviral effect [30]. Interestingly and consistent with an effect of ribavirin that would not be purely antiviral, animals treated with ribavirin had a median survival of 20.5 days, as compared to 10 days in animals left untreated or receiving 150 mg/kg/day favipiravir.

To what extent these promising effects translate to clinical setting? Here as well we compared our results with those obtained in Ebola infection, where patients received high doses of favipiravir [31]. The residual drug concentrations of 46.1 or 25.9 $\mu$g.mL$^{-1}$ obtained respectively at 2 and 4 days after treatment initiation are more than $10 \times EC_{50}$, and may therefore be sufficient to generate a high antiviral efficacy. Although the pharmacokinetics of favipiravir is

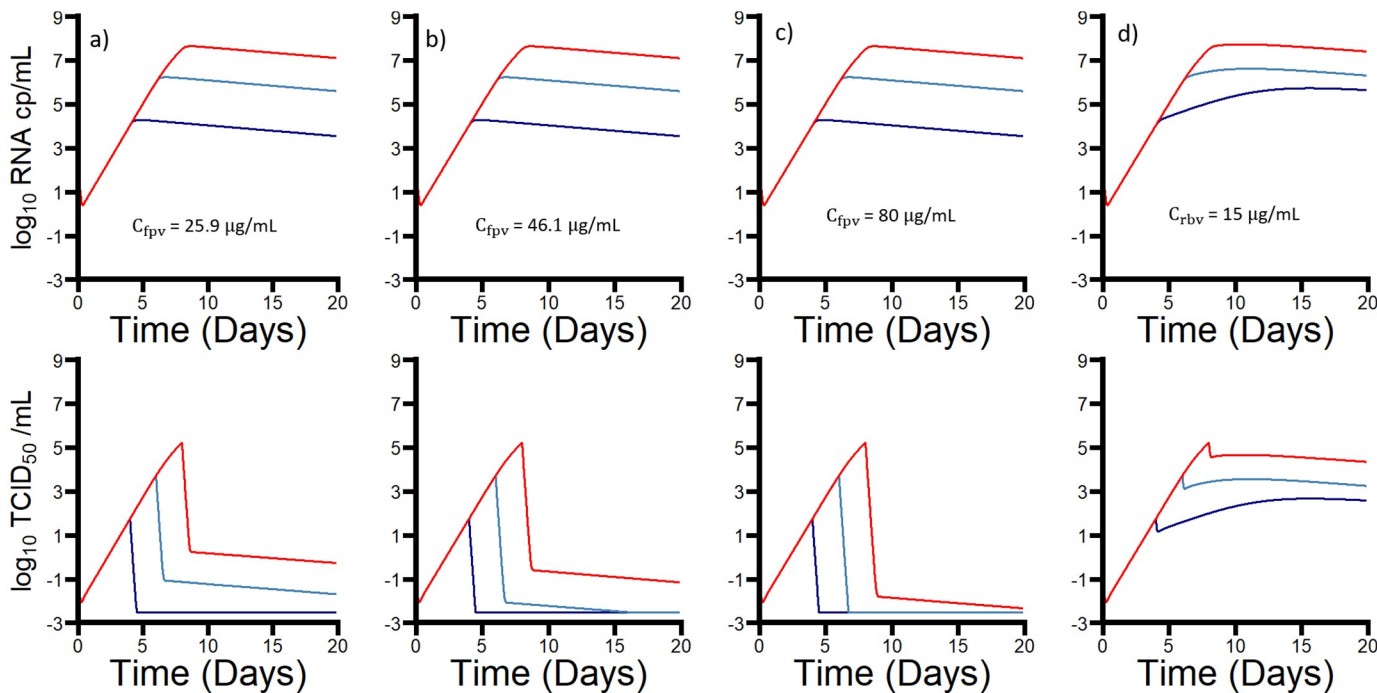

**Fig 4. Mutagenesis model simulations.** Constant favipiravir plasmatic concentration (a-c) or constant ribavirin plasmatic concentration (d). Treatment started at D4 (navy blue), D6 (blue), or D8 (red).

challenging and that some reductions in drug concentrations may occur over time [25] these results suggest that favipiravir could have a strong antiviral efficacy against Lassa virus. As discussed by we and others, antiviral treatment needs to be initiated before most target cells have been infected (i.e., before peak viral load) to have a large efficacy [8, 24, 27, 32]. By reducing peak viral load, treatment will in addition reduce the antigen stimulation and reduce the risk of a dysregulated immune responses. In clinical setting, the implication of these findings is treatment should be initiated as early as possible [33] and before peak viral load. This is thus a limitation for the translation of our results to human. However, with these levels of efficacy it is possible that viral load may be sufficiently reduced to accelerate the time to viral negativity, as suggested by our simulations (Fig 4). Importantly, the clinical efficacy of ribavirin against Lassa virus has never been demonstrated in clinical trials. Here our results show that ribavirin has a genuine antiviral efficacy; at the doses recommended by WHO guidelines (1000 mg every 6 h, [26, 34]), ribavirin may reduce virus infectivity by $\approx$ 80%. Given the other viral dynamic parameters, this efficacy may not be sufficient to drive the virus to error catastrophe (see Fig 4D). However it may still be relevant to reduce viral growth, lower inflammation and increase survival [8]. It is noteworthy that the mechanism of action of favipiravir may lead to a strong disconnection between the viral load kinetics, as measured by PCR, and the kinetics of infectious virus, measured by $TCID_{50}$. In that respect, the analysis of the viral dynamic should rely on both viral dynamics to precisely measure drug efficacy.

Several limitations need to be kept in mind. First, we assumed that LASV infection occurred only in one compartment. Although this may raise criticism [35], modeling multicompartmental infection will require data that do not exist at this time. We took the liver size as a proxy for the number of infected cells, which is supported by necropsy sampling showing that the liver was the organ having the highest level of viral replication (S8 Fig). Second we used a simple "cure boundary", based on concepts developed in other curable viral infections.

Although this hypothesis could well recapitulate the kinetics observed, the precise estimation of this cure boundary will require specifically designed experiment, such as studies involving shorter and repeated cure of treatments. In fact the existence of this cure boundary may be questioned, and may be more likely a virological control, as low levels of infectious virus were still detected in the liver and the cerebellum of "cured" animals long after treatment cessation (S8 Fig). Finally, and unlike what was done in Ebola infections, our model did not include an adaptive immune response [8]. This is due to the fact that we did not have any data on T or B-cell dynamics. Here our model focused primarily on the innate immune response. Following previous findings in Ebola virus [8], our model assumed an effect of the immune response in rendering cells refractory to infection, that could reflect the effects of IFN$\alpha$. We also considered several alternative models where an immune compartment could reduce viral infectivity, viral production, or increase the loss rate of free virus and infected cells, and verified the necessity of this immune response as well the presence of an effector compartment (S2 Text), but all alternative models showed deteriorated statistical criteria. Our prediction of the importance of IFN$\alpha$ is in line with *in vitro* [36] and in NHP [37] findings. Finally, and unlike what was done in Ebola infections, we did not incorporate an adaptive immune response in the model. Although IgG and IgM may be detectable 15 days after infection [37], it is noteworthy that the viral load concentrations did not show a rapid decline after peak viral load. Even surviving animals with undetectable viral titers at all times had detectable viral load until 40 days after infection, suggesting a modest role of the adaptive immune response in viral clearance. This immune effector is also known to reduce viral production, however favipiravir and ribavirin acting on the same target probably rendered this mode of action impossible to properly evaluate. Further viral dynamics after peak was characterized by a rather slow decline, suggesting that the adaptive immune response was limited, including in animals treated with high dose favipiravir.

To summarize our results provide the first description of viral dynamics in NHPs infected with Lassa virus. They provide target drug concentrations for ribavirin and favipiravir and strongly suggest that these drugs could be effective in reducing the proportion of infectious virus at dosing regimens relevant in humans.

## Supporting information

**S1 Fig. Pharmacokinetic models and concentrations predictions of treatments considered.**
a). Favipiravir model. b). Ribavirin model.
(TIF)

**S2 Fig. Model building procedure.** Steps of model selection.
(TIFF)

**S3 Fig. BIC of models tested throughout model building.** A-D) Step 1. A) FPV and RBV blocking production. B) FPV mutagen, RBV blocking production. C) RBV mutagen, FPV blocking production. D) FPV and RBV mutagens. E) Step 2-3. Mutagenesis model after random effect selection.
(TIF)

**S4 Fig. Predictions of titers/RNA at peak RNA viral load.** Top: Production blockage model. Bottom: Mutagenesis model. Yellow bars represent observed medians of each group.
(TIF)

**S5 Fig. Empirical Bayes estimates of parameter $\mu$.** EBEs were obtained using the production blockage model. $^{*}$ = p < 0.05. Horizontal lines represent medians by group.
(TIF)

**S6 Fig. Evolution of target cells TC, productive infected cells I2, refractory cells R and immune effector F.** Black curve: animals receiving vehicle treatment; green: animals treated with RBV; blue: animals treated with FPV 150 mg/kg/day; red: animals treated with FPV 300 mg/kg/day.
(TIF)

**S7 Fig. Parameters estimates distributions by model averaging.** Top: Production blockage model. Bottom: mutagenesis model.
(TIF)

**S8 Fig. Post-mortem RNA viral load by organ.** Black: Untreated. Green: Treated by ribavirin. Blue: Treated by low dose favipiravir. Red: Treated by high dose favipiravir.
(TIF)

**S1 Table. Pharmacokinetic parameters of favipiravir and ribavirin.** Values are estimated population values.
(PDF)

**S2 Table. BIC Comparison between modes of action of the immune response.** FPV and RBV mutagen agents, $\kappa = 1$.
(PDF)

**S3 Table. F compartment assessment.** Refractory model with FPV and RBV mutagen agents, $\kappa = 1$.
(PDF)

**S1 Text. Favipiravir pharmacokinetic model.** Equations for the pharmacokinetic model of favipiravir described in the litterature.
(PDF)

**S2 Text. Supplementary models equations.** Equations of models tested for model selection.
(PDF)

## Acknowledgments

We thank Sylvain Baize (Pasteur Institute) for critical reading of an initial version of the manuscript.

## Author Contributions

**Formal analysis:** Guillaume Lingas.

**Investigation:** Kyle Rosenke, David Safronetz.

**Writing – original draft:** Guillaume Lingas.

**Writing – review & editing:** Jérémie Guedj.

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
