## [Decision Letter · Decision Letter 0]

20 May 2020

Dear Mr Lingas,

Thank you very much for submitting your manuscript "Lassa viral dynamics in non-human primates treated with favipiravir or ribavirin" for consideration at PLOS Computational Biology.

As with all papers reviewed by the journal, your manuscript was reviewed by members of the editorial board and by several independent reviewers. In light of the reviews (below this email), we would like to invite the resubmission of a significantly-revised version that takes into account the reviewers' comments.

This is a very nicely analyzed study of lassa virus dynamics. The main concern is to improve the exposition along the lines indicated by reviewer 1.

We cannot make any decision about publication until we have seen the revised manuscript and your response to the reviewers' comments. Your revised manuscript is also likely to be sent to reviewers for further evaluation.

Sincerely,

Rustom Antia

Associate Editor

PLOS Computational Biology

Rob De Boer

Deputy Editor

PLOS Computational Biology

This is a very nicely analyzed study of lassa virus dynamics. The main concern is to improve the exposition along the lines indicated by reviewer 1.

Reviewer's Responses to Questions

**Comments to the Authors:**

Reviewer #1: The paper describes the first mathematical model of Lassa virus infection and uses NHP data to determine the roles of favipiravis and ribavirin in reducing the virus load and inducing viral cure. While I believe the study is important and has translational potential, the model development, the data fitting, and the model results need a more clear presentation. I will detail that below:

- It is very difficult to determine which results are based on the data and which are based on the model. This is especially true since the authors use the word dynamic to describe the mathematical model but also Table 1 which I believe is based on data. Same question regarding figures 1 and 2,are they based on data? Clarify.

- The mathematical model is quite confusing. Why are you using X for multiplication in some terms but not in others? Why is there an influx a*D/vol*e^(-a*c_t)t in the infectious virus population? What is D? Why is a accounted twice in that term? Define all these parameters. You call vol=V in the paper, chose one notation.

-Eclipse phase is I1 not I2.

-presentation of the density dependent terms needs to be closer to the standard. Define Phi and name k=Hill coefficient. Talk about what having a higher k means in terms of cooperative effects.

- The PK model in the figure S1 implies a more complicated form. Have you fitted it to PK data? Explain more what it does, how did you obtain the magnitude and the decay, and move it to the main manuscript, especially since it forms the basis for functions C.

- Why do you assume the interferon renders target cells refractory rather than blocking infection and production as modeled in influenza?

- Are you looking at four or five models? Define them better and separate the results.

- It is not clear which parameters are fixed and which are fitted to the data. It is not until table 2 is referenced that one can infer it.

- Are you fitting censored data?

- Explain your assumption regarding data fitting. For example: say you are using the first or second version of equation (10). What do you mean by deteriorated fitting criterion? What do you mean by 'at odds with data'. Show what you get or not get. Quantify it. Talk goodness of fit.

- Can you really fit both Theta and Phi?

- Remove the subtitle in the discussion.

- In the discussion you compare your results to those from Ebola where favipiravir increased virus diversity. You did not look into that scenario in Lassa, so you cannot make that comparison.

Reviewer #2: In this article, the authors develop viral dynamics models and analyze data from non-human primates to assess the efficacy and mode of action of the drugs favipiravir and ribavirin against Lassa fever. They assume that the drugs could either prevent viral replication or render the virions produced non-infectious, based on previous reports on the modes of action of these drugs. They analyze data from 24 primates administered either favipiravir or ribavirin post challenge and compare with a no treatment group. They also compare their results with independent observations on Ebola virus infection, for which favipiravir was used as treatment. Importantly, the measurements include total viral load as well as infectious virus titre, allowing estimation of the effect of the drugs on the infectiousness of the virus. The model also incorporates the overall drug pharmacokinetics based on the dosing protocols employed. Mixed effects analysis is used for fitting and parameter estimation. The authors find that favipiravir and ribavirin both work against Lassa fever by rendering the virions non-infectious and in a dose dependent manner. Further, favipiravir does so better than ribavirin and better than it performed against Ebola. They predict based on these findings the favipiravir dosage that would be suitable for treatment of Lassa fever in humans.

Overall, the study is carefully done. The results are important. The modeling approach is built on previous formalisms and demonstrating its applicability to in vivo data for Lassa fever for the first time and the ability to make predictions of the suitable dosing in humans make it valuable. The paper is well written. I have a few comments for the authors to consider.

Comments:

1. The authors explicitly incorporate an innate immune response in the model, which protects cells from getting infected. While this effect is expected in most viral infection settings, it is rarely used explicitly in models. The authors should thus explain in some detail why they chose to do so. It may also help to state explicitly what components of the innate immune response are important in the present setting. NK cells and interferon, for instance, could have opposite effects on the outcomes of infection (e.g., see https://www.pnas.org/content/116/35/17393)

2. The authors argue that the timing of the start of treatment is crucial for achieving viral clearance. Does this have to do with the viral load rising with time and thus treatment having to suppress a much larger viral pool when started late or is it something more fundamental that is changing with time, like the innate immune response? In other words, can I think of outcomes of treatment being dependent on the viral load at the start of treatment regardless of when the treatment is initiated or does the treatment initiation time have an explicit role? Could the authors clarify and explain? Also, if the timing is really the issue, how could the time of infection be estimated?

3. Lethal mutagenesis and error catastrophe have a threshold like dependence on the mutation rate. As a first approximation, the error threshold is ~1/L, where L is the length of the viral genome (see the work of Manfred Eigen, Peter Schuster and colleagues). The authors do argue that of the two dosages of favipiravir used, the lower does not trigger an error catastrophe and the higher does. Are there in vitro estimates of the mutation rate induced by favipiravir as a function of its concentration that they could use to justify this claim at least based on the ~1/L value? If not, could they propose further studies that would be needed to establish the error threshold for Lassa virus and the associated favipiravir dosage? There seem to be few examples of mutagenic drugs inducing error catastrophe in vivo. It would be remarkable therefore if the authors could demonstrate this. This may also have clinical implications. Because of the threshold behavior, suboptimal dosing may lead to an increase in the mutation rate and hence viral diversity that may favor immune escape or trigger drug resistance. Ensuring that the threshold is crossed may thus be important.

Minor comments:

1. Line 90: Should I2 be I¬1?

2. In Eqs 7 and 8, the Hill coefficient is set to 1 for ribavirin but a much higher value for favipiravir. Is this just for simplicity or is there evidence of the coefficient being 1 for ribavirin?

3. In Eqs. 9 and 10, it may help to distinguish between the two modes of drug action using different symbols or subscripts. Right now, they are both epsilon.

4. Line 150: The viral clearance rates in tissues and plasma are set to the same value, saying that the same has been done with Zika and Ebola. It may help to add a line saying why this works with Zika and Ebola.

5. Line 191: Ribavirin concentration is assumed fixed. Ribavirin concentration, however, is known to show massive accumulation with time, which the authors too recognize in their pharmacokinetics model. Should this pharmacokinetics not be used for predictions, especially since the dosing for Lassa fever may not last longer than the timescale of the concentration build up?

6. Should drug resistance be a cause of worry at all? Is it reported? Perhaps a comment about it in the discussion would help.

7. In Fig. 1, there seems to be a long tail with the low favipiravir dosage? Is that just a sampling issue or is it systematic?

**Have all data underlying the figures and results presented in the manuscript been provided?**

Reviewer #1: No: The code for data fitting and the individual subject data are not provided.

Reviewer #2: None

PLOS authors have the option to publish the peer review history of their article (what does this mean?). If published, this will include your full peer review and any attached files.

Reviewer #1: No

Reviewer #2: No
---

## [Decision Letter · Decision Letter 1]

13 Nov 2020

Dear Mr Lingas,

We are pleased to inform you that your manuscript 'Lassa viral dynamics in non-human primates treated with favipiravir or ribavirin' has been provisionally accepted for publication in PLOS Computational Biology.

Best regards,

Rustom Antia

Associate Editor

PLOS Computational Biology

Rob De Boer

Deputy Editor

PLOS Computational Biology

Reviewer's Responses to Questions

**Comments to the Authors:**

Reviewer #1: The authors have addressed my comments. The paper is much improved and ready to be published.

Reviewer #2: The authors have addressed my concerns. No further comments.

**Have all data underlying the figures and results presented in the manuscript been provided?**

Reviewer #1: Yes

Reviewer #2: None

PLOS authors have the option to publish the peer review history of their article (what does this mean?). If published, this will include your full peer review and any attached files.

Reviewer #1: No

Reviewer #2: No

---

## [Editor Report · Acceptance letter]

30 Dec 2020

PCOMPBIOL-D-20-00583R1 

Lassa viral dynamics in non-human primates treated with favipiravir or ribavirin

Dear Dr Lingas,

I am pleased to inform you that your manuscript has been formally accepted for publication in PLOS Computational Biology. Your manuscript is now with our production department and you will be notified of the publication date in due course.

With kind regards,

Livia Horvath
